# A simple theory for quantum quenches in the ANNNI model

Jacob H. Robertson[1]⋆, Riccardo Senese[1] and Fabian H. L. Essler[1]

**1** The Rudolf Peierls Centre for Theoretical Physics, Oxford University, Oxford OX1 3NP, UK
⋆ jacob.robertson@physics.ox.ac.uk

## Abstract

In a recent numerical study [1] it was shown that signatures of proximate quantum critical points can be observed at early and intermediate times after certain quantum quenches. Said work focused mainly on the case of the axial next-nearest neighbour Ising (ANNNI) model. Here we construct a simple time-dependent mean-field theory that allows us to obtain a quantitatively accurate description of these quenches at short times and a surprisingly good approximation to the thermalization dynamics at late times. Our approach provides a simple framework for understanding the reported numerical results as well as fundamental limitations on detecting quantum critical points through quench dynamics. We moreover explain the origin of the peculiar oscillatory behaviour seen in various observables as arising from the formation of a long-lived bound state.

# 1 Introduction

Quantum phase transitions (QPT) provide a key framework for our understanding of equilibrium phases of correlated quantum matter [2]. More recently physical properties in the vicinity of quantum critical points in out-of-equilibrium settings have been investigated theoretically [3–5] and in ultra-cold atom experiments [6–9]. An interesting question that has been raised is whether it is possible to detect the location of QPTs, and associated physical properties, through the dynamics at short and intermediate times after a quantum quench [1,10–14]. In Ref. [1] Haldar et al proposed a set of generalized susceptibilities that quantify the sensitivity of the time evolution and stationary values of local observables to changes in the quench protocol. Based on numerical studies in the axial next-nearest neighbour Ising model (ANNNI) the authors concluded that such susceptibilities can indeed provide signatures of a proximate QPT not only in the stationary regime but already at short/intermediate times. An important question is how general this approach is, and what its limitations are. In order to address these issues we show that the findings of Ref. [1] for the ANNNI model can be understood in terms of a simple (time-dependent) mean-field theory. This approach provides a clear insight into the window of applicability of any approach using generalized susceptibilities to search for the location of critical points. En route we clarify the origin of interesting oscillatory behaviours of local observables observed in Ref. [1].

The outline of this paper is as follows. In Section 2 we introduce the ANNNI model and describe the quench protocol we consider. In Section 3 we then construct a mean-field description of the stationary state under the assumption that the system thermalizes. In Section 4 we construct a time-dependent self-consistent mean-field description of the time evolution. Within this approximation the density matrix is Gaussian at all times and Wick's theorem may be employed to calculate any correlation function. This method is expected to be quantitatively accurate for short times as long as the initial state is itself Gaussian. In Section 5 we show that non-equal time correlation functions are easily accessible with this method and use it to compute the transverse component of the generalized dynamical structure factor following a quench in the ANNNI, demonstrating that this object contains information about the spectrum of the post-quench Hamiltonian.

# 2 Definition of the model and quench protocol

The ANNNI model is a well studied non-integrable model with competing interactions, see e.g. [15–18]. The model consists of a transverse-field Ising model with an additional next-nearest neighbour Ising exchange, which we take to have the opposite sign to the nearest-neighbour Ising interaction

$$H(h, \kappa) = -J \sum_i^L \sigma_i^x \sigma_{i+1}^x - h \sum_i \sigma_i^z + \kappa \sum_i^L \sigma_i^x \sigma_{i+2}^x \,. \tag{1}$$

Here $\sigma_j^\alpha$ are the usual Pauli matrices on sites $j$ of a ring of circumference $L$. The Hamiltonian (1) can be mapped to a model of spinless lattice fermions by means of a Jordan-Wigner transformation [19]. As we adopt periodic boundary conditions for the spins the fermions must obey either anti-periodic (Neveu-Schwarz) or periodic (Ramond) boundary conditions depending on whether the fermion number is even or odd, see e.g. Appendix A of [20]. For our purposes it is sufficient to work in the Neveu-Schwarz sector for even system sizes $L$. The

Hamiltonian then reads

$$H(h,\kappa) = -J \sum_j \left( c_j^\dagger c_{j+1} + c_j^\dagger c_{j+1}^\dagger + \text{h.c.} \right) + \kappa \sum_j (c_j^\dagger c_{j+2} + c_j^\dagger c_{j+2}^\dagger + \text{h.c.}) + 2h \sum_j c_j^\dagger c_j$$

$$+ 2\kappa \sum_j \left( c_j c_{j+1}^\dagger c_{j+1} c_{j+2}^\dagger - c_j^\dagger c_{j+1}^\dagger c_{j+1} c_{j+2}^\dagger + \text{h.c.} \right) . \tag{2}$$

The next-nearest neighbour spin-spin interaction is seen to give rise to a quartic interaction amongst the fermions, making the model non-integrable. The Hamiltonian (1) has a global $\mathbb{Z}_2 \otimes \mathbb{Z}_2$ symmetry corresponding to rotations around the $z$-axis by $\pi$ – which is broken spontaneously in the ferromagnetic phase – and site parity $\sigma_n^\alpha \mapsto \sigma_{-n}^\alpha$. The latter remains unbroken in the situations we consider and enforces $t_{ij} \equiv \langle c_i^\dagger c_j \rangle = t_{ji} \in \mathbb{R}$ (see Appendix A), while the former translates into fermion number parity.

   The ground state phase diagram of the ANNNI model for $\kappa < 0.5$ is shown in Fig. 1 [16,21–23]. At $\kappa = 0$ the model (1) reduces to the transverse field Ising model (TFIM) and is exactly

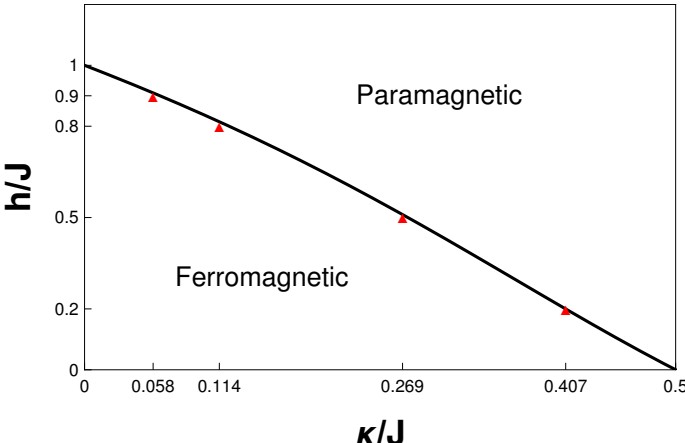

Figure 1: Ground state phase diagram of the ANNNI model for $0 < \kappa/J < 1/2$ - the solid curve is the boundary obtained by second order perturbation theory (3), red triangles indicate the critical points found by our self-consistent mean-field theory at select fields. There exist other phases at $\kappa > J/2$.

solvable as it is quadratic in fermions [2,19]. For $\kappa > 0$ a second order phase transition in the Ising universality class separates a ferromagnetically ordered phase from a paramagnetic one. For $\kappa < J/2$ and small values of $h$ the locus of the critical line can be determined by second order perturbation theory, which yields [15]

$$J - 2\kappa_c = h_c - \frac{1}{2J} \frac{\kappa_c h_c^2}{J - \kappa_c} . \tag{3}$$

In terms of the spins the transition is characterized by the order parameter $\langle \sigma_j^x \rangle$ taking a non-zero value in the ferromagnetic phase. In terms of the fermions this is a non-local (string) operator and the transition is topological [24]. Our analysis of quench dynamics close to quantum critical points in one dimension therefore pertains to both topological transitions and conventional transitions with local order parameters. Moreover, our mean-field analysis developed below is exact along the line $\kappa = 0$ and correctly accounts for the symmetry and critical exponents of the Ising transition for $\kappa > 0$. Hence it is expected to give a quantitatively accurate description of the ANNNI model in the region $h \approx J$ and $\kappa \approx 0$.

In what follows we consider quantum quenches from initial thermal states of the TFIM with transverse field $h_i$ and inverse temperature $\beta$, i.e. our initial density matrix is

$$\rho(t=0) = \frac{\exp\left(-\beta H(h_i, 0)\right)}{\text{Tr} \exp\left(-\beta H(h_i, 0)\right)} \, . \tag{4}$$

Including thermal states at finite temperatures rather than only ground states is useful as it allows us to tune the energy density of the stationary state reached at late times in a simple manner. We then consider the time evolution induced by the ANNNI Hamiltonian $H(h_f, \kappa)$, i.e.

$$\rho(t > 0) = e^{-iH(h_f, \kappa)t} \rho(t=0) e^{iH(h_f, \kappa)t} \, . \tag{5}$$

We will restrict ourselves to the case $h_i = h_f \equiv h$ and quenches with $\kappa < J/2$. To simplify notations we also set $J = 1$. As the ANNNI model is non-integrable when both $h$ and $\kappa$ are non-zero we expect the model to thermalize [4,25], i.e. in the thermodynamic limit the system should locally relax to a thermal stationary state described by an effective temperature that is set by the energy density of the initial state

$$e_0 = \lim_{L \to \infty} \frac{1}{L} \text{Tr}\left(\rho(t=0) H(h_f, \kappa)\right). \tag{6}$$

In our setup the correlation length typically starts off small as a result of a large pre-quench gap, while at late times the system settles into a thermal state at a low effective temperature in the vicinity of a quantum critical point. Hence the correlation length in the stationary state is typically much larger than in the initial state. Intuitively therefore the physics should be that of a system whose correlation length grows following the quench.

## 3  Mean-field theory for the Stationary State

Since the ANNNI model is believed to thermalize and has no local conservation laws other than the total energy, we expect local observables $\mathcal{O}$ to reach their Gibbs ensemble values at late times after a quantum quench

$$\langle \mathcal{O} \rangle(t) \xrightarrow{t=\infty} Z^{-1} \text{Tr}[e^{-\beta_f H(h_f, \kappa)} \mathcal{O}] \, . \tag{7}$$

Here $Z$ is the partition function and $\beta_f$ the inverse effective temperature, set by the initial energy density (6) generated by the quench protocol. For sufficiently small values of $\kappa$ this thermal state should be amenable to a description in terms of a simple self-consistent mean-field theory of spinless fermions

$$Z^{-1} \text{Tr}[e^{-\beta_f H} \mathcal{O}] \approx Z_{\text{MFT}}^{-1} \text{Tr}[e^{-\beta_{\text{MFT}} H_{\text{MFT}}} \mathcal{O}] \, , \tag{8}$$

where

$$H_{\text{MFT}} = \sum_i \sum_{a=0}^{2} \left\{ J_{\text{Eff}}^{(a)} (c_i^\dagger c_{i+a} + \text{hc}) + (\Delta_{\text{Eff}}^{(a)} c_i^\dagger c_{i+a}^\dagger + \text{hc}) \right\} + E_0 \, . \tag{9}$$

This mean-field theory is the result of requiring that Wick's theorem holds, or equivalently that higher cumulants vanish. The effective couplings $J_{\text{Eff}}^{(a)}$ and $\Delta_{\text{Eff}}^{(a)}$ and the constant $E_0$ are generated by decoupling the quartic interaction terms self-consistently via

$$ABCD \mapsto \langle AB \rangle_{\text{MFT}} CD + AB \langle CD \rangle_{\text{MFT}} - \langle AB \rangle_{\text{MFT}} \langle CD \rangle_{\text{MFT}} + \text{all other Wick contractions} \, , \tag{10}$$

where

$$\langle \mathcal{O} \rangle_{\text{MFT}} \equiv Z_{\text{MFT}}^{-1} \text{Tr}[e^{-\beta_{\text{MFT}} H_{\text{MFT}}} \mathcal{O}] . \tag{11}$$

Defining the (self-consistent) expectation values

$$
\begin{aligned}
t_a &\equiv \langle c_j^\dagger c_{j+a} \rangle_{\text{MFT}} , \quad a = 0, 1, 2 , \\
\Delta_b &\equiv \langle c_j^\dagger c_{j+b}^\dagger \rangle_{\text{MFT}} , \quad b = 1, 2 ,
\end{aligned}
\tag{12}
$$

we have

$$
\begin{aligned}
J_{\text{eff}}^{(0)} &= h - 2\kappa(t_2 + \text{Re}\,\Delta_2) , \\
J_{\text{eff}}^{(1)} &= -(J - 4\kappa(t_1 + \text{Re}\,\Delta_1)) , \quad \Delta_{\text{eff}}^{(1)} = -(J - 4\kappa(t_1 + \Delta_1^*)) , \\
J_{\text{eff}}^{(2)} &= \kappa(1 - 2t_0) , \quad\quad\quad\quad\quad\quad \Delta_{\text{eff}}^{(2)} = \kappa(1 - 2t_0) , \\
E_0 &= -hL - 4L\kappa(|\Delta_1|^2 + t_1^2 - t_0 t_2 + 2\,\text{Re}\,\Delta_1 t_1 - \text{Re}\,\Delta_2 t_0) .
\end{aligned}
\tag{13}
$$

In order to fully specify our self-consistent mean-field theory we require the self-consistent values of the five mean-fields as well as the value of the inverse effective temperature $\beta_{\text{MFT}}$, which is fixed by the condition that the energy density in the stationary state is the same as in the initial state (6), i.e.

$$e_0 = \lim_{L \to \infty} \frac{\langle H_{\text{MFT}} \rangle_{\text{MFT}}}{L} . \tag{14}$$

The various self-consistency equations are most easily solved in momentum space. As stated above it is sufficient to work in the Neveu-Schwarz sector for even system sizes $L$, so that

$$c_k \equiv \frac{1}{\sqrt{L}} \sum_m e^{ikm} c_m , \quad k \in \left\{ 2\pi \frac{n + 1/2}{L} , \; n = -\frac{L}{2}, \dots, \frac{L}{2} - 1 \right\} . \tag{15}$$

The mean-field Hamiltonian then becomes

$$
\begin{aligned}
H_{\text{MFT}} &= \sum_{k>0} A_k (c_k^\dagger c_k - c_{-k}^\dagger c_{-k}) + i B_k (c_k^\dagger c_{-k}^\dagger) - i B_k^* (c_{-k} c_k) + \text{const} , \\
A_k &= 2 \sum_{a=0}^{2} J_{\text{eff}}^{(a)} \cos ak , \quad B_k = 2 \sum_{a=1}^{2} \Delta_{\text{eff}}^{(a)} \sin ak .
\end{aligned}
\tag{16}
$$

We remark that in equilibrium not just the $t_a$ but also the $\Delta_b$ are in fact real despite the absence of a unitary symmetry enforcing this, see Appendix A. This in turn makes it possible to diagonalize the Hamiltonian by a one-parameter Bogoliubov transformation

$$b_\kappa(k) = \cos \frac{\theta_\kappa(k)}{2} c(k) - i \sin \frac{\theta_\kappa(k)}{2} c^\dagger(-k) , \quad e^{i\theta_\kappa(k)} = \frac{A_k - i B_k}{\sqrt{A_k^2 + B_k^2}} , \tag{17}$$

which gives [1]

$$H_{\text{MFT}} = \sum_{k>0} \varepsilon_\kappa(k) b_\kappa^\dagger(k) b_\kappa(k) + \text{const} , \quad \varepsilon_\kappa(k) = \sqrt{A_k^2 + |B_k|^2}. \tag{18}$$

The self-consistency conditions on the mean-fields are given by calculating the expectation values using (11)

$$t_a = \frac{1}{L} \sum_k e^{-iak} \langle c_k^\dagger c_k \rangle_{\text{MFT}} = \frac{1}{L} \sum_{k>0} \cos ak \left( 1 - \cos \theta_\kappa(k) \tanh \frac{\beta_{\text{MFT}} \varepsilon_\kappa(k)}{2} \right) , \tag{19}$$

$$\Delta_a = \frac{1}{L} \sum_k e^{-iak} \langle c_k^\dagger c_{-k}^\dagger \rangle_{\text{MFT}} = \frac{1}{L} \sum_{k>0} \sin ak \sin \theta_\kappa(k) \tanh \frac{\beta_{\text{MFT}} \varepsilon_\kappa(k)}{2} , \tag{20}$$

---

[1] Here we write $|B_k|^2$ which gives the correct dispersion for complex $B_k$, as it will be out-of-equilibrium, although the form of the required canonical transformation in (17) will be more complicated.

while the equation fixing the effective temperature (6) takes the form

$$4\kappa\left((t_1+\Delta_1)^2-(t_0-1/2)(t_2+\Delta_2)\right)_{\kappa=0}+h-\frac{1}{L}\sum_{k>0}\varepsilon_{\kappa=0}(k)\tanh\frac{\beta_i\varepsilon_{\kappa=0}(k)}{2}$$

$$=E_0+J^{(0)}_{\text{Eff}}-\frac{1}{L}\sum_{k>0}\varepsilon_\kappa(k)\tanh\frac{\beta_{\text{MFT}}\varepsilon(k)}{2}. \tag{21}$$

The initial energy density given by the left hand side of (21) is a constant for fixed values of $\kappa, h$, however the right-hand side depends upon the values of the mean-fields and thus this equation must be solved self-consistently along with the other conditions on the mean-fields.

Eqs (19)-(21) need to be solved numerically, where the Bogoliubov angles are defined by Eq (17) and Eq (13). The solutions can be directly compared to numerical results obtained in Ref. [1] via a numerical linked cluster expansion [26, 27]. In Fig. 2 we plot the mean-field

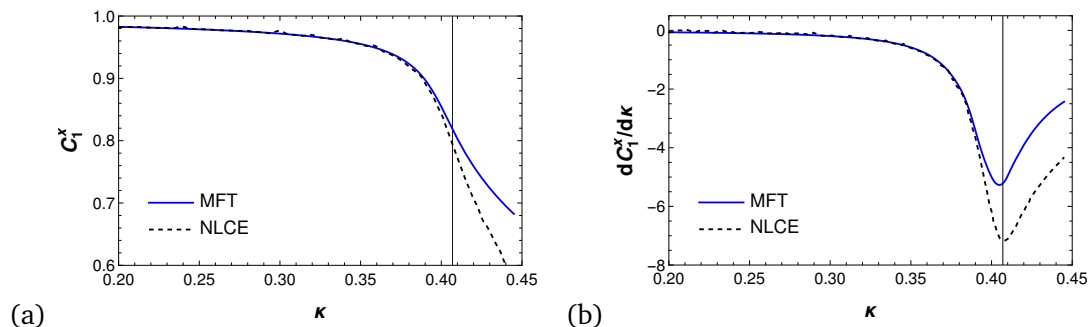

(a)                (b)

Figure 2: (a) $C_1^x = 2(t_1+\Delta_2)$ in the thermal state reached at late times after a quench from the TFIM ground state at $h = 0.2$ as a function of $\kappa$. The solid blue line is the result obtained from our self-consistent mean-field theory and the dashed black line shows numerical linked cluster expansion (NLCE) results extracted from [1]. (b) Same comparison as (a) but for $\chi_1 = \partial_\kappa C_1^x(\kappa)$. The vertical lines indicate $\kappa_c$.

results for the longitudinal nearest-neighbour correlator

$$C_1^x \equiv \langle\sigma_i^x\sigma_{i+1}^x\rangle = 2(t_1+\text{Re}\,\Delta_1)\,, \tag{22}$$

in the (thermal) steady state following a quench from the ground state of the TFIM with $h = 0.2$ along with the susceptibility $dC_1^x/d\kappa$ defined using an ensemble of quenches. We see that the agreement of our mean-field analysis with the numerical results of Ref. [1] is excellent up to fairly large values of $\kappa$. We observe similarly good agreement with the transverse magnetization $m^z \equiv \langle\sigma_j^z\rangle$ and the next-nearest neighbour longitudinal correlator $C_2^x \equiv \langle\sigma_i^x\sigma_{i+2}^x\rangle$. In Fig. 3 we compare the self-consistent inverse temperature $\beta_{\text{MFT}}$ to numerical results of Ref. [1]. We observe excellent agreement essentially over the full range of $\kappa$ considered.

Given the good agreement with state-of-the-art numerical results we conclude that our self-consistent fermionic mean-field theory provides a good description of the steady state reached at late times after the quenches considered.

## 3.1 Scaling regime at finite energy densities

The key objective of Ref. [1] was to establish that quantum quenches can be used to locate the positions of quantum phase transitions in some parameter space. An important question is to what extent the observed signatures are indeed associated with the scaling behaviour induced by the proximate quantum critical point. To answer this question by purely numerical

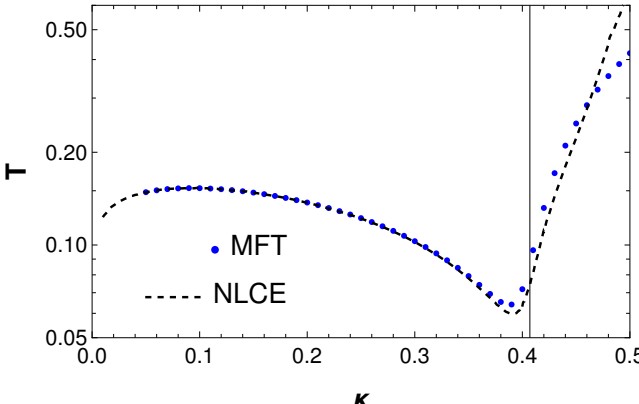

Figure 3: Comparison of $T = \beta_{\mathrm{MFT}}^{-1}$ to effective temperatures reported in [1]. The dashed black curve shows the NLCE results reported in Fig. 8 of [1], while the blue data points are the values found by our self-consistent mean-field theory. The vertical line indicates $\kappa_c$.

methods would require the analysis of the long-distance behaviour of correlation functions or entanglement entropies of large sub-systems, in order to ascertain whether they display scaling behaviour characteristic of the proximate quantum critical point. Our mean-field theory gives us a much simpler way of answering this question: as the field theory describing the quantum critical point is a gapless relativistic Majorana fermion the scaling regime extends at most to energies per particle at which the mean-field dispersion is still to a good approximation linear. These considerations set an energy cut-off for the field theory. In Fig. 4 we plot the mean-field dispersion relation (18) and compare it to the respective effective temperatures. We see from

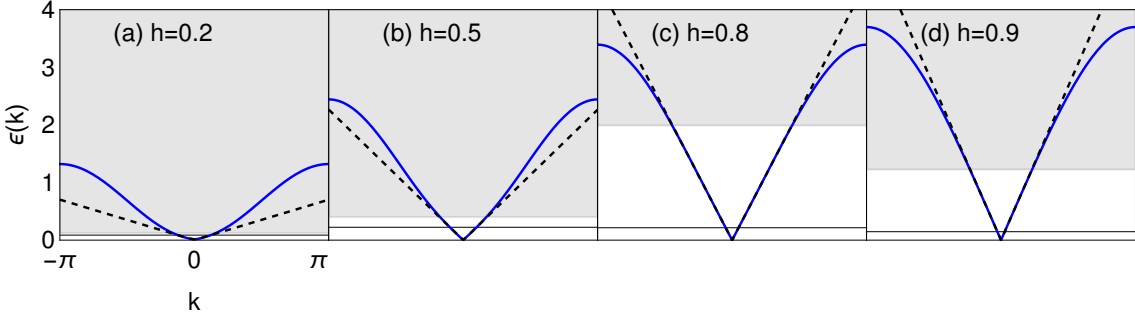

Figure 4: Effective dispersion relations in the steady state following a quench with (a) $h = 0.2$, $\kappa = 0.407 \approx \kappa_c$, (b) $h = 0.5$, $\kappa = 0.269 \approx \kappa_c$, (c) $h = 0.8$, $\kappa = 0.114 \approx \kappa_c$, (d) $h = 0.9$, $\kappa = 0.058 \approx \kappa_c$ . The black horizontal line is the effective temperature $T = \beta_{\mathrm{MFT}}^{-1}$. The dashed black line is a fit to $\varepsilon_{\mathrm{fit}}(k) = \sqrt{\varepsilon_\kappa(0)^2 + v_{\mathrm{fit}}^2 k^2}$ and the gray shaded region indicates the regime of energy densities where spectral non-linearities become significant and corrections to scaling limit behaviour can no longer be expected to be negligible.

Fig. 4(c,d) that when $h$ is close to 1 and $\kappa$ small, the scale over which the dispersion is linear is much larger than the effective temperature. This implies that for these quenches the steady state is in fact in the scaling regime of the Ising transition and properties of the underlying quantum critical point are readily accessible.

By contrast in Fig. 4(a,b) we show the mean-field dispersion relation (18) in the steady state for quenches with small $h$ and large $\kappa$ can be fitted with a relativistic dispersion only for a small energy window. Here the scale over which the Majorana dispersion is linear is very

small and of the same order of magnitude as the effective temperature. This means that for these quenches the steady state is *outside* the scaling regime of the Ising transition, and so we can't actually glean any useful information about the underlying quantum critical point using quench dynamics.

We expect the fact that the cut-off decreases for smaller values of $h$ to be an accurate prediction of the mean-field theory presented here in light of the good agreement with the numerics seen in Fig. 2. The point that the energy density needs to be sufficiently below the cut-off scale of the quantum critical point one is trying to probe is of course both obvious and very general.

## 4    Self-consistent time-dependent mean-field theory (SCTDMFT)

Following Refs [28–34] we now turn to the dynamics after our quantum quenches in the framework of a self-consistent time-dependent Gaussian approximation. This amounts to considering time evolution with a time-dependent mean-field Hamiltonian

$$H_{\text{MFT}}(t) = \sum_i \sum_{a=0}^{2} \left\{ J_{\text{Eff}}^{(a)}(t)(c_i^\dagger c_{i+a} + \text{hc}) + (\Delta_{\text{Eff}}^{(a)}(t)c_i^\dagger c_{i+a}^\dagger + \text{hc}) \right\} + E_0(t) , \qquad (23)$$

where the time-dependent couplings are given by the time-dependent analogs of (13), i.e.

$$t_a(t) = \text{Tr}\left( \rho_{\text{MFT}}(t)c_j^\dagger c_{j+a} \right) , \quad a = 0, 1, 2 ,$$

$$\Delta_b(t) = \text{Tr}\left( \rho_{\text{MFT}}(t)c_j^\dagger c_{j+b}^\dagger \right) , \quad b = 1, 2 ,$$

$$\rho_{\text{MFT}}(t) = \left\{ \mathcal{T} e^{-i\int_0^t H_{\text{MFT}}(t')dt'} \right\} \rho(t=0) \left\{ \mathcal{T} e^{-i\int_0^t H_{\text{MFT}}(t')dt'} \right\}^\dagger . \qquad (24)$$

Here $\mathcal{T}$ denotes time ordering; the initial density matrix $\rho(t=0)$ (4) is by construction Gaussian and concomitantly so is $\rho_{\text{MFT}}(t)$. This is the essence of the SCTDMFT, which by construction is expected to work best at short times. This is because it is based on the assumption that all higher cumulants vanish, which is strictly true at time $t = 0$. At short times the higher cumulants will become non-zero, but their growth is expected to be slow for small $\kappa$. At late times SCTDMFT is not expected to work well in general [35,36] and in some models is known to describe relaxation towards a "prethermalization plateau" [37–39] rather than thermalization. However, as we will see, it works reasonably well even at late times for some of the quenches considered here.

As a consequence of the translation invariance of the problem the time-evolved Gaussian density matrix $\rho_{\text{MFT}}(t)$ is fully characterised by the two momentum space two-point averages

$$\tilde{t}_k(t) = \text{Tr}\left( \rho_{\text{MFT}}(t)\, c_k^\dagger c_k \right) , \qquad \tilde{\Delta}_k(t) = \text{Tr}\left( \rho_{\text{MFT}}(t)\, c_k^\dagger c_{-k}^\dagger \right). \qquad (25)$$

The self-consistent equations of motion for these $k$ space two-point functions can be obtained using the Heisenberg equations of motion associated to the (now time-dependent) analog of the momentum space Hamiltonian (16). The result is

$$\frac{d\tilde{\Delta}_k(t)}{dt} = 2iA_k(t)\tilde{\Delta}_k(t) + B_k^*\left[ 1 - 2\tilde{t}_k(t) \right]$$

$$\frac{d\tilde{t}_k(t)}{dt} = 2\,\text{Re}\left( B_k(t)\tilde{\Delta}_k(t) \right) , \qquad (26)$$

where

$$A_k = 2\sum_{a=0}^{2} J_{\text{eff}}^{(a)}(t)\cos ak , \qquad B_k = 2\sum_{b=1}^{2} \Delta_{\text{eff}}^{(b)}(t)\sin ak . \qquad (27)$$

We now integrate the equations (26) using a second-order midpoint scheme with a timestep of $10^{-3}$, which we choose to ensure that the mean-fields are converged with respect to the timestep. At each timestep we must update the real space mean-fields $t_a$ and $\Delta_b$ using $\tilde{t}_k$ and $\tilde{\Delta}_k$

$$t_a = \frac{1}{L} \sum_k \tilde{t}_k(t) e^{-ika} , \qquad \Delta_b = \frac{1}{L} \sum_k \tilde{\Delta}_k(t) e^{-ikb} . \qquad (28)$$

Physical quantities such as spin-spin correlation functions can then be calculated in terms of (sums of products of) the fermionic two-point functions.

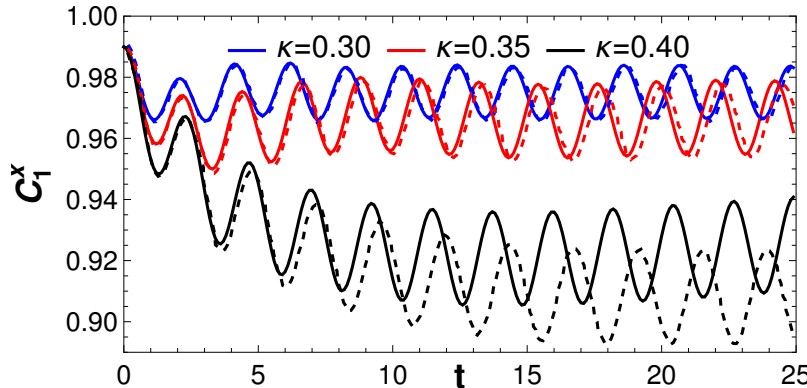

Figure 5: Comparison of SCTDMFT results for $C_1^x(t)$ to iTEBD results taken from [1] for a quench from the ground state at $h = 0.2, \kappa = 0$ to $\kappa > 0$. Here the solid lines are SCTDMFT results for $L = 2000$ and the dashed lines in the respective color are iTEBD. The agreement is seen to be very good except for near the critical point ($\kappa_c \approx 0.407$).

## 4.1 Short and intermediate-time behaviour of local correlation functions

In Fig. 5 we compare the results of the above SCTDMFT approximation to iTEBD results taken from [1], which are believed to be essentially numerically exact. For small values of $\kappa$ compared to the critical value $\kappa_c$ we find excellent agreement over the entire time range accessible to iTEBD. For larger values of $\kappa$ the agreement is still very good at short times, but gets worse at late times.

While Ref. [1] focused on spin correlations, the time evolution of the fermionic two-point functions is of interest as well, in particular in relation to the question of detecting topological transitions by quench dynamics. In Fig. 6 we present results obtained by SCTDMFT for $t_1(t)$ and $\text{Re}(\Delta_1(t))$ following quenches from the ground state of $H(h, \kappa = 0)$ with $h = 0.2, 0.8$ to $\kappa = 0.05, 0.20$. We observe the following:

- For quenches with small transverse fields $h$ there are persistent oscillations around a constant value, which is in good agreement with the corresponding expectation value after thermalization.

- For quenches at large fields $h$ there are no long-lived oscillations. Instead the expectation values relax to stationary values that differ from the ones predicted by thermalization by an amount that scales at $\mathcal{O}(\kappa^2)$. This is expected by virtue of the perturbative nature of the mean-field approximation.

An explanation of the oscillatory behaviour is provided below in section 4.3.

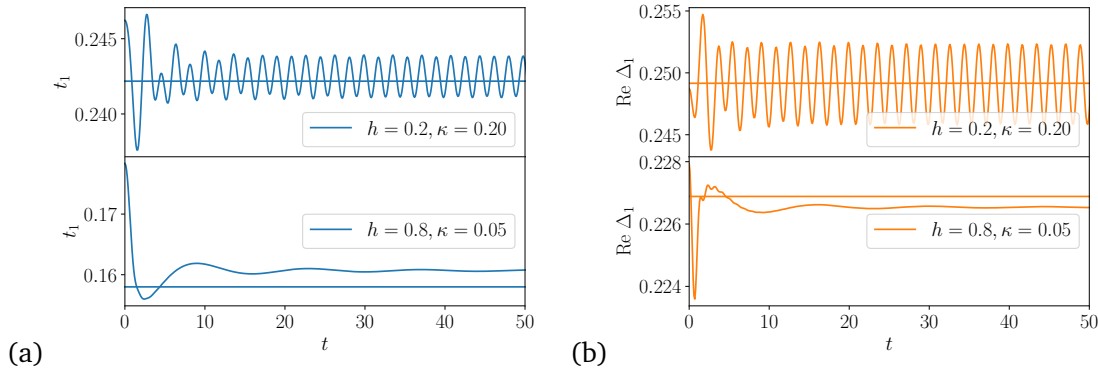

(a)                                                                        (b)

Figure 6: Nearest neighbour fermion two-point functions $t_1(t)$, $\mathrm{Re}\,\Delta_1(t)$ after quenches from the ground state of $H(h, \kappa = 0)$ with $h = 0.2$ and $h = 0.8$ to $H(h, \kappa)$. Horizontal lines indicate the stationary values found in Section 3.

As suggested in [1], a signature of the proximate quantum phase transition can be obtained by processing data for the expectation value of a local observable for an ensemble of quenches at a fixed time $t$ after the quench. In Fig. 7 we show results for $C_1^x(t)$ and $dC_1^x(t)/d\kappa$ for an ensemble of quenches starting in the ground state of $H(h, \kappa = 0)$ and quenching to $H(h, \kappa)$ for $h = 0.2, 0.8$ and a wide range of $\kappa$ values.

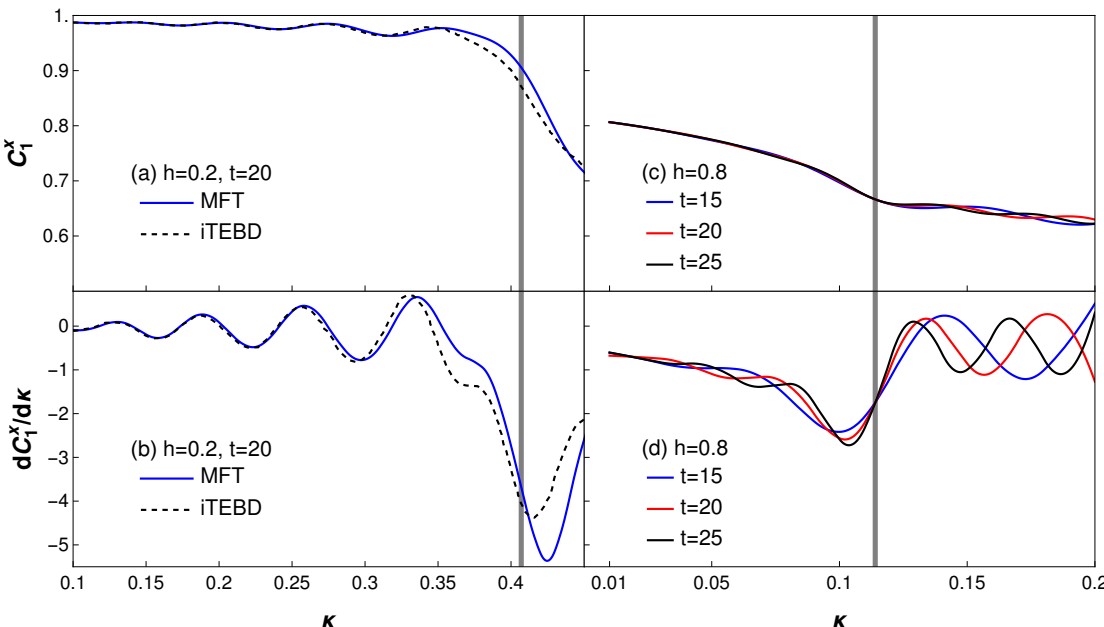

Figure 7: Performing quenches from $H(h, 0)$ to $H(h, \kappa)$ we build a picture of observables as a function of final $\kappa$. (a-b) Comparison with iTEBD data taken from [1] for $h = 0.2$ ($\kappa_c \approx 0.407$, indicated by thick gray line). (c-d) Equivalent calculation at $h = 0.8$ ($\kappa_c \approx 0.114$). All quenches done starting from the ground state for system size $L = 2000$.

In Fig. 7(a-b) we find very good agreement between our SCTDMFT results and the iTEBD simulations of Ref. [1] for $h = 0.2$ and in Fig. 7(c-d) we show the results for $h = 0.8$. The generalized susceptibility $dC_1^x/d\kappa$ in Fig. 7(b,d) shows a strong dip even at the relatively early time $t = 20$ around the critical value $\kappa_c$. Intuitively one expects that the reason for this strong response to the varying post-quench parameters is that the correlation length at time $t = 20$ is

already large and the system "feels" the proximity of the QPT; this implies a large correlation length and consequently a strong linear response of the system, reflected in the dips in generalized susceptibilities. We return to this point in the next section where, in Fig. 9, we extract correlation lengths for the non-equilibrium state of the system following the quench for $h = 0.8$ and find that the correlation length has grown from $\xi \approx 1.9$ at $t = 0$ to $\xi \approx 12$ at $t = 20$. Conversely, in cases where the correlation length is short we do not expect the susceptibility to be large. This is indeed the case for small values of $\kappa$ in Fig. 7. In Fig. 8 we show the time

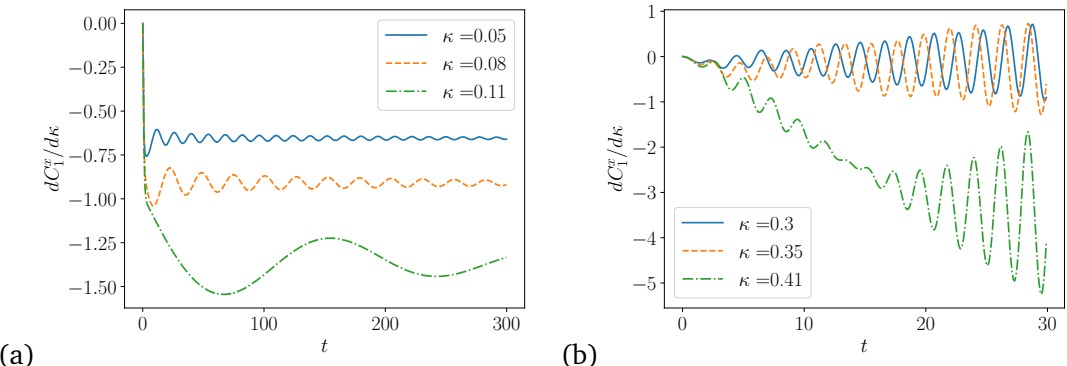

(a)                                                                                    (b)

Figure 8: Short time dynamics of the generalized susceptibility for quenches from an initial thermal state with $\beta = 2.0$ and (a) $h = 0.8$ ($\kappa_c \approx 0.114$) and (b) $h = 0.2$ ($\kappa_c \approx 0.407$) on a system with $L = 2000$.

evolution of the generalized susceptibilities. Fig. 8 shows , for two values of $h$, quench data for various $\kappa$, including near the critical value $\kappa_c$. For $\kappa$ far from $\kappa_c$ we observe a quick relaxation to a plateau, whilst for $\kappa$ close to the QPT we observe a longer relaxation time. Fig. 8(b) features growing oscillations due a 'beat' phenomenon when numerically differentiating between the different quench data with slightly different persistent oscillation frequencies.

## 4.2 Growth of the correlation length in time

As we have noted above, the correlation length grows in time for many of the quenches we consider. To show this explicitly we focus on the connected order-parameter two-point function

$$C_{c,\ell}^x(t) = \underbrace{\text{Tr}\left[\rho_{\text{MFT}}(t)\,\sigma_n^x \sigma_{n+\ell}^x\right]}_{C_\ell^x(t)} - \left(\text{Tr}\left[\rho_{\text{MFT}}(t)\,\sigma_n^x\right]\right)^2, \tag{29}$$

as it is easier to extract a correlation length for than $\sigma_j^z$. Since the order parameter expectation value is itself difficult to calculate even in the TFIM [40, 41] we follow Ref. [42] in using the Lieb-Robinson bound [43] to express the connected correlator as

$$C_{c,\ell}^x(t) = C_\ell^x(t) - C_R^x(t), \quad R \gg v_{\max} t, \tag{30}$$

where $v_{\max}$ is the Lieb-Robinson velocity. In our self-consistent mean-field approximation we can use Wick's theorem to express $C_\ell^x(t)$ as a block-Toeplitz Pfaffian [44]

$$C_\ell^x(t) = \text{Pf}\begin{pmatrix} G_0(t) & G_1(t) & \dots & G_{\ell-1}(t) \\ -G_1^T(t) & \ddots & \ddots & \vdots \\ \vdots & \ddots & \ddots & \vdots \\ -G_{\ell-1}^T(t) & \dots & \dots & G_0(t) \end{pmatrix}, \tag{31}$$

where

$$G_n(t) = 2 \begin{pmatrix} i\,\mathrm{Im}\,\Delta_n(t) & \mathrm{Re}(t_{1-n}(t) + \Delta_{1+n}(t)) - \frac{1}{2}\delta_{0,n+1} \\ -\mathrm{Re}(t_{1-n}(t) + \Delta_{1-n}(t)) + \frac{1}{2}\delta_{0,1-n} & i\,\mathrm{Im}\,\Delta_n(t) \end{pmatrix}. \quad (32)$$

We note that if we replace the time-dependent Gaussian density matrix by a thermal equilibrium state Eq (31) reduces to a determinant because $\Delta_n \in \mathbb{R}$.

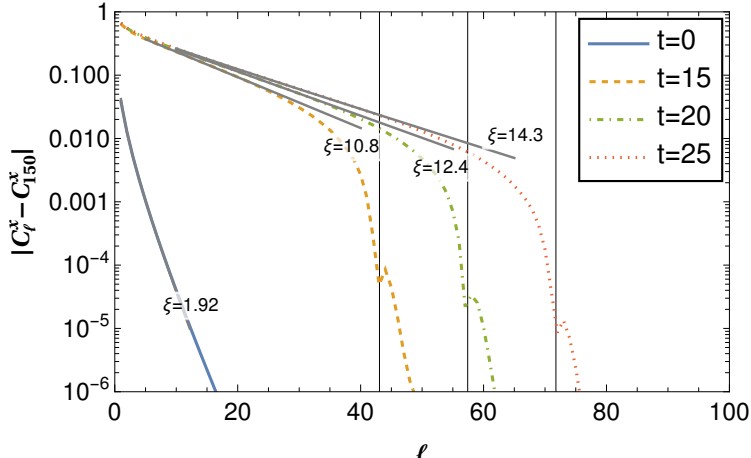

Figure 9: Connected order-parameter two-point function $C^x_{c,\ell}(t)$ for a quench from the ground state of the TFIM at $h = 0.8$ to the ANNNI with $h = 0.8, \kappa = 0.11$ ($\kappa_c \approx 0.114$). Vertical lines indicate the lightcone distance at $t = 15, 20, 25$ using the maximal group velocity of the effective dispersion in the steady state. Gray lines indicate fits to functions of the form $C^x_{c,\mathrm{fit}} = a\ell^{-\nu}\exp(-\ell/\xi)$ where $\xi$ is the fitted correlation length.

In Fig. 9 we show the connected order-parameter two-point function for a quench from the ground state of the TFIM with $h = 0.8$ and turning on next nearest neighbour interactions of strength $\kappa = 0.11$. In the initial state the connected correlator displays exponential decay with a correlation length $\xi(0) \approx 1.9$. Extracting correlation lengths at $t > 0$ is complicated by the fact that the connected correlator for outside the "light-cone" remains unchanged and we are therefore restricted to separations $\ell < 2v_{\mathrm{max}}t$, where $v_{\mathrm{max}}$ is the maximal propagation velocity [4,45,46]. On the other hand, in order to extract a correlation length $\xi(t)$ we require that $\ell \gg \xi(t)$. This causes us to be unable to convincingly fit correlation lengths for short times (other than $t = 0$ which is an equilibrium state by design), although we obtain relatively good fits to the exponential behaviour at times $t \geq 20$ which show the correlation length has grown to about $\xi(25) \approx 14.3$.

## 4.3 Oscillations in the low energy-density regime

A striking feature seen in Figs. 5, 6, 8 are the high-frequency oscillations in local observables for quenches at reasonably small $h$ which do not appear to decay in time in the mean-field theory. These do not occur in quenches in the TFIM and hence seem to be a result of fermion interactions. We stress that these oscillations were previously observed in the iTEBD simulations of Ref. [1] and are not an artifact of the mean-field approximation. Importantly they are observed in quenches that result in small energy densities compared to the fermion gap, which puts us in a regime where we are dealing with the non-equilibrium dynamics of a very dilute gas of fermions. This suggests that these oscillations could be related to the formation of long-lived bound states of (pairs of) fermions, *cf.* Refs [47–51]. A simple limiting case in

which this bound state formation can be seen is $h = 0$. Here excitations are (highly degenerate) domain-wall states, whilst the antiferromagnetic next-nearest neighbour term partially lifts this degeneracy by introducing an energy penalty of $4\kappa$ when the domain-walls are on exactly neighbouring bonds. That is, at $h = 0$ the next-nearest neighbour interaction produces a spin-flip (anti-)bound state. In order to investigate the possibility of these bound states persisting to the non-zero values of $h$ we consider we have determined the spectrum of low-lying excitations of the ANNNI model by exact diagonalization using the QuSpin [52] package on $L = 24$ sites. These results provide useful information for physical properties at finite energy densities that are small compared to the excitation gap over the ground state. As in the ferro-

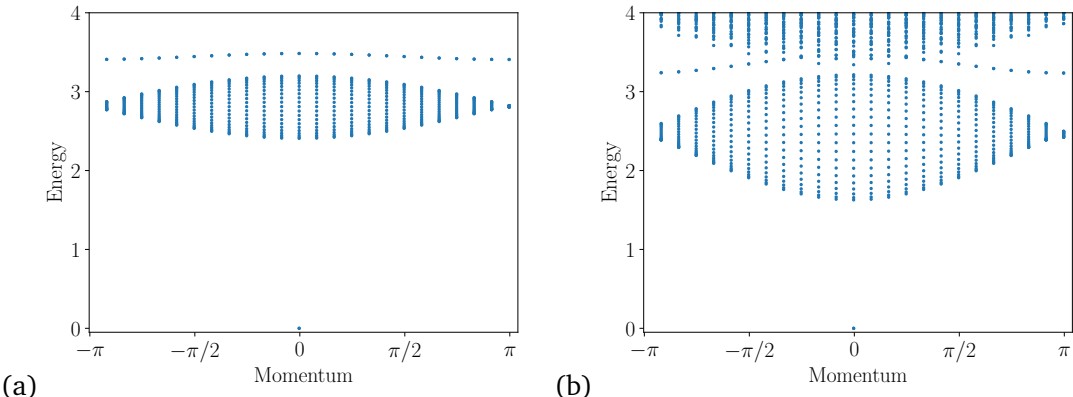

(a)                                                    (b)

Figure 10: Spectrum of the ANNNI Hamiltonian for (a) $h = 0.1$, $\kappa = 0.15$ and (b) $h = 0.2$, $\kappa = 0.2$ from exact diagonalisation using QuSpin [52] on $L = 24$ sites. As physical states have even fermion parity, the lowest excited states are the two domain-wall continuum and a sharp bosonic mode corresponding to the anti-bound state. For $h = 0.2$, $\kappa = 0.2$ the four-particle continuum is low enough in energy to be visible on this scale.

magnetic phase of the TFIM the lowest excitations can then be thought of as a continuum of pairs of ferromagnetic domain-walls. This is indeed observed in the exact diagonalization results in Fig. 10. In addition we observe a bosonic bound state of two domain-walls that occurs at energies above the two domain-wall continuum. With regards to the oscillations observed in local observables after some of our quenches we note the following:

- The bound state energy at $k = 0$ agrees with the oscillation frequency observed after the quantum quenches.

- For reasonably large values of $h$ the bound state ceases to exist around $k = 0$. It can be seen from a Lehmann representation that only excited states with $k = 0$ contribute to the dynamics when performing quenches from translationally invariant states as we do here. As such this is consistent with the fact that when we perform quenches with larger $h$ we do not see persistent oscillations.

An important caveat is that in the quench set-up we are dealing with there is a small, but finite, energy density above the ground state and thus in the thermodynamic limit the system is in fact at an energy infinitely above what is pictured in Fig. 10. There the bound states always "sit" on top of multi domain-wall excitations and are not expected to be stable. However, as the density of domain-walls is very small the life-time of the bound state can be very large compared to the time scale we observe in our quenches. We believe that this is indeed the case.

A rough estimate of the decay time of the bound states can be obtained by thinking in the quasiparticle picture described above. If there were truly a single bound state then energy

and momentum conservation would prevent it from decaying, however the decay is allowed due a background density of domain walls that the bound state may scatter from. A semi-classical approach to compute the scattering time is to introduce the mean-free-path of the domain-walls

$$\lambda_{\text{mfp}} = \frac{E_g}{\varepsilon} \, , \tag{33}$$

where $\varepsilon$ is the energy density *relative to the ground state after the quench* and $E_g$ the quasiparticle gap. If the mean-free-path is larger than the system size $\lambda_{\text{mfp}} > L$, then the state has in expectation fewer than one quasi-particle in the entire system and the system does not require a many-body description and the bound states will have nothing to scatter from. Even for thermodynamically large systems however if we consider times less than

$$2v_{\text{max}}t \lesssim \lambda_{\text{mfp}} \, , \tag{34}$$

where $v_{\text{max}}$ is the Lieb-Robinson velocity of the domain-wall excitations, we may consider the bound state quasiparticles as having little interaction with the domain-wall background. We now estimate all the relevant quantities in the case of interest. The post-quench energy

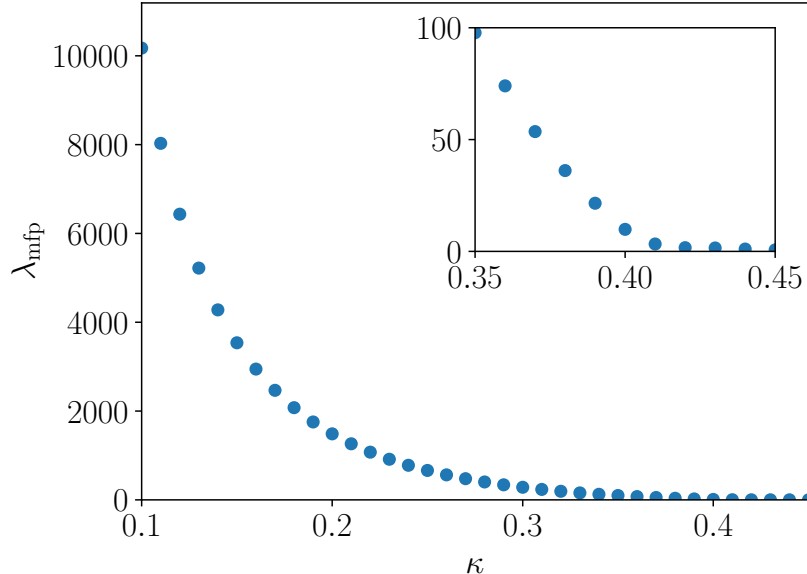

Figure 11: Mean free path of the quasiparticles generated by quantum quenches from the TFIM ground state at transverse field $h = 0.2$ to the ANNNI model with $0.1 < \kappa < 0.45$ ($\kappa_c \approx= 0.407$).

density $e_0$ defined in (14) may be calculated using Wick's theorem. The energy density $\varepsilon$ appearing in Eq (33) is however not the $e_0$ of (14) but rather one must subtract the ground state energy density of the ANNNI, which is not known analytically. We estimate the latter by exact diagonalization for $L = 18$ sites, for which it is essentially converged. The resulting mean-free-path for quenches from the ground state of the TFIM with $h = 0.2$ to the ANNNI model with $0.1 < \kappa < 0.45$ is shown in Fig. 11. We see that for these quenches the mean free path is extremely large unless $\kappa$ is very close to the QPT. The time range accessible to us in our SCTDMFT analysis is limited by finite-size effects, which strongly influence observables after the traversal time $L/(2v_{\text{max}})$ [4, 53, 54]. To access very late times without encountering finite-size effects therefore requires larger system sizes and more memory. In order to test whether or not the oscillations eventually decay in mean-field theory we instead change our initial density matrix in a way that reduces the mean free path, e.g. for a quench with $h = 0.1$

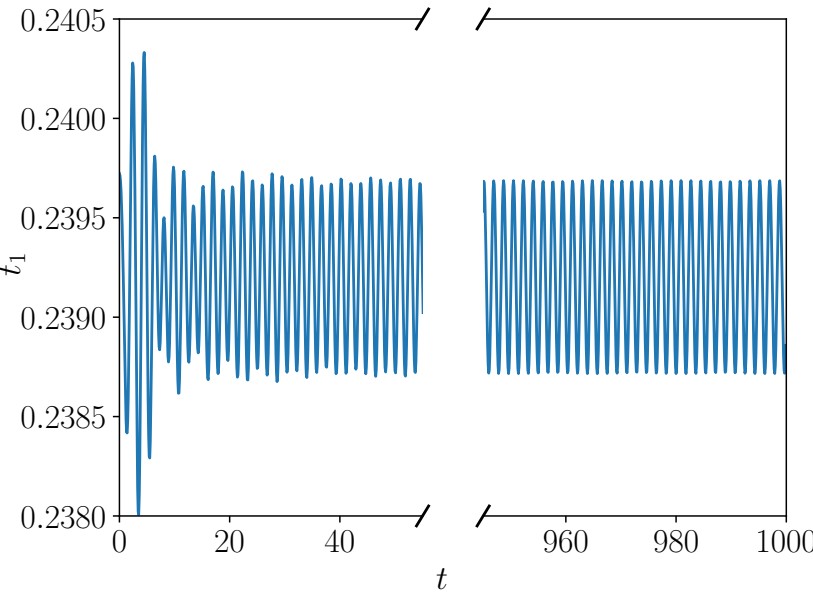

Figure 12: Time evolution of the mean field $t_1$ following a quench from $\beta = 2.0$, $h = 0.1, \kappa = 0.15$.

and $\kappa = 0.15$ from an initial temperature $\beta = 2.0$, we estimate that the mean free path should be roughly 50 sites and the scattering time about $t_s \sim 56$, see Table (1). Nonetheless there is no visible damping in the mean-field theory up to very late times ($t = 10^3$), see Fig. 12. We

| $e_0(\beta = 2.0)$ | $e_{GS}(h = 0.1, \kappa = 0.15)$ | $\varepsilon$ | $2E_g$ | $\lambda_{\mathrm{mfp}}$ | $v_{\mathrm{max}}$ | $t_s$ |
|---|---|---|---|---|---|---|
| -0.82739 | -0.85295 | 0.02556 | 2.410 | 47.14 | 0.4187 | 56.29 |

Table 1: Postquench energy density $e_0$ obtained from Eq (14), ground state energy density $e_{GS}$ and two particle gap estimated with ED on $L = 20$ sites. Lieb-Robinson velocity is estimated as the maximal group velocity for the dispersion $\epsilon_\kappa(k)$ given in Eq (18) using the values of the mean-fields at $t = 100$.

conclude that in SCTDMFT the oscillations are *undamped* while we expect in an exact theory they would decay.

## 5 Non-equal time correlation functions

A natural question is whether the existence of a bound state can be detected more directly in the quench setup. One proposal in the literature is to use certain Fourier transforms of equal-time correlation functions [55, 56], but these do not provide useful insights in our case. In thermal equilibrium it is well established that dynamical response functions give detailed information about the particle content of the theory. An obvious question then is to what extent their non-equilibrium analogs can be used to do the same. In order to address this question we now determine certain non-equal time correlation functions in our SCTDMFT. We do not attempt to address the problem of calculating non-equal time two-point functions of the order parameter, as this is difficult even for the transverse field Ising chain itself [41, 57]. In MFT the Heisenberg equations of motion for the fermion operators $c_k$ are linear

$$\frac{\mathrm{d}}{\mathrm{d}t} c_k(t) = i[H_{\mathrm{MFT}}(t), c_k(t)] = -iA_k(t)c_k(t) + B_k c^\dagger_{-k}(t) , \tag{35}$$

and can be solved by a time-dependent Bogoliubov transformation

$$c_k(t) = \alpha_k(t)c_k(0) + \beta_k(t)c^\dagger_{-k}(0) \,, \tag{36}$$

where the time-dependent coefficients are solutions to

$$\frac{\mathrm{d}\alpha_k(t)}{\mathrm{d}t} = -iA_k(t)\alpha_k(t) + B_k(t)\beta^*_{-k}(t) \,, \qquad \frac{\mathrm{d}\beta_k(t)}{\mathrm{d}t} = -iA_k(t)\beta_k(t) + B_k(t)\alpha^*_{-k}(t) \,. \tag{37}$$

As we are dealing with a Gaussian theory all non-equal time correlation functions are then expressible in terms of the two non-equal time Green's functions given by

$$\begin{aligned}
G_k(t,t') = \langle c^\dagger_k(t)c_k(t') \rangle &= \alpha^*_k(t)\alpha_k(t')f_k + \alpha^*_k(t)\beta_k(t')g_k \\
&\quad + \beta^*_k(t)\alpha_k(t')g^*_k + \beta^*_k(t)\beta_k(t')(1 - f_{-k}) = G_{-k}(t,t') \,, \tag{38}
\end{aligned}$$

$$\begin{aligned}
\tilde{G}_k(t,t') = \langle c^\dagger_k(t)c^\dagger_{-k}(t') \rangle &= \alpha^*_k(t)\alpha^*_{-k}(t')g_k + \alpha^*_k(t)\beta^*_{-k}(t')f_{-k} \\
&\quad + \beta^*_k(t)\alpha^*_{-k}(t')(1 - f_k) + \beta^*_k(t)\beta^*_{-k}(t')g^* = -\tilde{G}_{-k}(t,t') \,, \tag{39}
\end{aligned}$$

where expectation values are always taken with respect to $\rho(t = 0)$, i.e. $\langle \mathcal{O} \rangle = \mathrm{Tr}[\rho(t = 0)\mathcal{O}]$. The final equalities hold due to the parity symmetry and $f_k$, $g_k$ encode the initial conditions

$$f_k = G_k(0,0), \qquad g_k = \tilde{G}_k(0,0) \,. \tag{40}$$

As an example of the use of these formulas we consider the non-equilibrium analog of the density response function

$$\chi_{\rho\rho}(r,t,t') = \frac{1}{L^2} \sum_{k_1,\ldots k_4} e^{i(k_1 - k_2)r} \langle [c^\dagger_{k_1}(t)c_{k_2}(t), c^\dagger_{k_3}(t')c_{k_4}(t')] \rangle \,. \tag{41}$$

After Fourier transforming in the spatial co-ordinate this takes the following form in SCTDMFT

$$\begin{aligned}
\tilde{\chi}(q,t,t') = \frac{1}{L} \sum_k \Big\{ &\tilde{G}_k(t,t')\tilde{G}^*_{k-q}(t',t) - \tilde{G}_k(t',t)\tilde{G}^*_{k-q}(t,t') \\
&+ G_k(t,t')\Big(\alpha^*_{k-q}(t')\alpha_{k-q}(t) + \beta^*_{k-q}(t')\beta_{k-q}(t)\Big) \\
&- \Big(\alpha^*_k(t)\alpha_k(t') + \beta^*_k(t)\beta_k(t')\Big) G_{k-q}(t',t) \Big\} \,. \tag{42}
\end{aligned}$$

We note that $\chi(q,t,t')$ is in principle measurable via linear-response measurements, see Appendix B. Employing a Lehmann representation suggests that spectral properties of the post-quench Hamiltonian should be inferrable by taking appropriate "Fourier transforms" in time. In practice we consider

$$\chi_{t_f}(q,\omega) = \int_0^{t_f} dt' \, \tilde{\chi}(q,t_f,t') \, e^{i\omega t'}. \tag{43}$$

The imaginary part of this generalized dynamical susceptibility is shown in Fig. 13 for a quench from $\kappa = 0$ to $\kappa = 0.15$ and initial temperature $\beta = 1.0$. We can clearly identify the continuum of two domain-wall excitations but there is no evidence for a bound state above it. In order to capture the latter one has to go beyond the SCTDMFT.

# 6   Conclusion

We have formulated both equilibrium (at finite energy density) and time-dependent mean-field descriptions for quantum quenches in the ANNNI model starting from a Gaussian state.

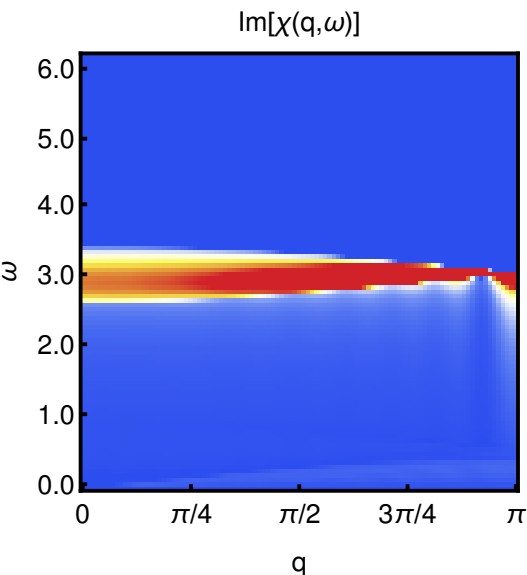

Figure 13: Out-of-equilibrium density-density susceptibility calculated for the mean-field theory with $L = 200, h = 0.1, \kappa = 0.15, \beta = 1.0$

We first used this to compute properties of the expected stationary state following a quantum quench, assuming that the system looks thermal again at late times and then used the time-dependent formulation to probe the approach to stationarity. Comparisons in both the stationary and time-dependent cases with the numerical results of Ref. [1] show that this simple description is surprisingly accurate even for large next-nearest neighbour interactions close to the critical value. Importantly it fully reproduces the signatures of the equilibrium phase transition previously found numerically. Our approach makes it clear that the observed signatures are associated with the growth of the correlation length following a quantum quench and sheds light on the applicability of this mechanism for detecting quantum phase transitions in general. Our theory is based on a fermionic description with a topological transition and so it is clear that topological as well as conventional transitions may be detected in this manner. Moreover, we give an explanation for a potentially puzzling feature of the real time dynamics, namely long-lived oscillations, by showing that the oscillation frequency is the mass of a bound state in the interacting theory.

Finally, we showed that the time-dependent mean-field approach used here is capable of calculating non-equal time correlation functions, however it is unable to capture the bound state produced by the quartic interaction in the theory.

## Acknowledgements

This work was supported by the EPSRC under grant EP/S020527/1. We are grateful to A. Das for drawing our attention to Ref. [1] and helpful discussions.

## A    Reality of certain mean-fields

When evaluating our self-consistent mean-fields we observe that some of them are real. In this appendix we explain why this is the case, beginning with a clarification of the site parity $\sigma_j^\alpha \mapsto \sigma_{-j}^\alpha$. This does not act on the fermions as $c_j \mapsto c_{-j}$ due to the presence of the Jordan-

Wigner string. The simplest way to deduce the effect of site parity in the fermion basis is to look at the action of site parity on fermion bilinears, which can be simply related to spin operators without semi-infinite strings. In particular, we consider the following spin bilinears of definite parity

$$
\begin{aligned}
A &= \sigma_i^x \sigma_{i+1}^x \,, \\
B &= \sigma_i^y \sigma_{i+1}^y \,, \\
C^\pm &= \sigma_i^x \sigma_{i+1}^x \pm \sigma_i^y \sigma_{i+1}^y \,.
\end{aligned}
\tag{44}
$$

We then note that the fermionic bilinears can be decomposed in terms of these via

$$
\begin{aligned}
c_i^\dagger c_j &= \frac{1}{4}(A + B - iC^-) \,, \\
c_j^\dagger c_i &= \frac{1}{4}(A + B + iC^-) \,, \\
c_i^\dagger c_j^\dagger &= \frac{1}{4}(A - B - iC^+) \,, \\
c_i c_j &= \frac{1}{4}(-A + B - iC^+) \,.
\end{aligned}
\tag{45}
$$

We thus see that the action of site parity on the bilinears is to exchange $c_i^\dagger c_j$ with $c_j^\dagger c_i$ and therefore $t_{ij} = \langle c_i^\dagger c_j \rangle = t_{ji} \in \mathbb{R}$ as stated in the main text.

Additionally, the ANNNI Hamiltonian satisfies $H = H^* = H^T$ in both the spin and fermion bases. In particular, in the fermion basis $c_i c_j$ is also real. By the spectral theorem for real symmetric matrices we then know that the eigenvectors of $H$ are real in the same basis and so

$$
\langle c_i c_j \rangle_\beta = \frac{1}{Z(\beta)} \sum_n \langle E_n | c_i c_j | E_n \rangle e^{-\beta E_n} \in \mathbb{R}
\tag{46}
$$

is manifestly real in equilibrium. However, after the quench the corresponding time-evolved quantity becomes

$$
\langle c_i c_j \rangle_t = \frac{1}{Z(\beta)} \sum_{n,n',m'} e^{-\beta E_n^0} \langle E_n | E_{m'} \rangle \langle E_{m'} | c_i c_j | E_{n'} \rangle \langle E_{n'} | E_n \rangle e^{-it(E_{m'} - E_{n'})} \,,
\tag{47}
$$

where $E_n^0$ are the pre-quench energies and $E_{m'}$ the post-quench energies. Even if the post-quench Hamiltonian is also real and thus the post-quench energy eigenstates $|E_{n'}\rangle$ real, the phase factors will cause it to be generically complex. However, at very late times we would expect that the system would come back to equilibrium via these factors dephasing and so the correlation function should become real again at late times. Since $t_n$ are all real due to the site parity $\mathbb{Z}_2$ this implies that all effective couplings are real in equilibrium, and out of equilibrium the only complex one will be $\Delta_{\text{Eff}}^{(1)}(t)$.

## B  Linear response

In this appendix we summarize how to derive Kubo linear response relations after a quantum quench that occurs at time $t = 0$, see e.g. Ref. [58]. The Hamiltonian is of the form

$$
H(t) = \theta(-t) H_i + \theta(t) H_f + f(t) V \,,
\tag{48}
$$

where $\theta(t)$ is the Heaviside step function. If $f = 0$ this corresponds to a quench at $t = 0$. The linear response regime is when $f(t) \ll 1$ and for this to be genuinely non-equilibrium we require $f(t)$ to have support in the time period before the system thermalizes after the quench.

We work in an interaction picture such that $H = H_0 + f(t)V$, where $H_0$ is generally not free. The interaction picture states $|\psi(t)\rangle_I$ are defined by

$$|\psi(t)\rangle_I = e^{iH_0 t} U(t, t_0)|\psi(t_0)\rangle \,, \tag{49}$$

where $U(t, t_0)$ is the full time-evolution operator associated with $H(t)$, $|\psi(t_0)\rangle$ is the Schrödinger picture state at $t_0$ and $H_0$ is considered time independent by requiring, according to (48), that $t_0 \geq 0$. Consistently, in the interaction picture the general operator $\mathcal{O}$ evolves in time as

$$\mathcal{O}_I(t) = e^{iH_0 t} \mathcal{O} e^{-iH_0 t} \,. \tag{50}$$

The time-evolution according to $H(t)$ of the expectation value of $\mathcal{O}$ in the state defined at $t_0$ by $\rho(t_0) = |\psi(t_0)\rangle\langle\psi(t_0)|$ can be expressed in the interaction picture as

$$\mathrm{Tr}(\rho(t)\mathcal{O}) = \mathrm{Tr}(\rho_I(t)\mathcal{O}_I(t)) \approx \mathrm{Tr}(\rho(t_0)\mathcal{O}_I(t)) - i\int_{t_0}^{t} f(t')\chi(t, t')\mathrm{d}t' \,, \tag{51}$$

where the susceptibility $\chi(t, t')$ is given by

$$\chi(t, t') \equiv \mathrm{Tr}\big(\rho(t_0)[\mathcal{O}_I(t), V_I(t')]\big) \,. \tag{52}$$

In the last step of (51) we have expressed $\rho_I(t) = |\psi(t)\rangle_{II}\langle\psi(t)|$ by the first two terms in its power series in the small function $f(t)$. Eq (51) is the usual linear response formula except that the time-translation invariance of the susceptibility is broken by the quench and hence $\chi(t, t')$ does not depend only on the time-difference $t - t'$.

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
