# Peer review of "A simple theory for quantum quenches in the ANNNI model"

_SciPost Physics_

## Round 1 · Referee Report · Anonymous (Referee 1) · 2023-3-14

Strengths

This work is timely, targeting an interesting open question in the field.

Weaknesses

None

Report

In this work the authors develop a mean-field theory for a paradigmatic quantum spin model in one dimension, the so-called ANNNI model. They apply their method to a dynamical problem, which has been recently studied from a purely numerical point of view. Specifically, this concerns signatures of quantum critical points visible in the dynamics after quantum quenches in certain quantum spin models. The authors convincingly and impressively show that their mean-field theory successfully captures the essential behaviors of the previously numerically obtained data. Most importantly, this work provides physical explanations for the numerical observations, such as the presence of a special type of bound state causing oscillatory dynamical behavior in various time-dependent observables.

This work is timely , targeting an interesting open question in the field. The manuscript is very well written and well structured. I therefore recommend publication of the article in Scipost.

I could imagine that the manuscript might be further improvable by accounting for the following point:

In Sec. 3 it is said that "... this thermal states should be amenable to a description in terms of a simple self-consistent mean-field theory...". The results shown later by the authors indeed demonstrate that. However, I wouldn't have guessed naturally that such a mean-field theory might be suitable for 1D systems. Is this because of the effect of temperature? In case the authors have further arguments for the applicability of mean-field theory for the considered model and/or in a more general context this might further improve the manuscript.

Overall, I recommend this manuscript for publication in Scipost.

Requested changes

None

---

## Round 1 · Referee Report · Anonymous (Referee 2) · 2023-3-15

Report

This manuscript studies quantum quenches in the ANNNI model using a self-consistent time-dependent mean field theory. Specifically, the analysis presented here is consistent with a recent exact numerical study [1] (citations [.] are from the manuscript) that show signatures of quantum critical points in certain observables at finite times. In addition, the analysis presented here points to an explanation of persistent oscillatory behavior in certain observables.

The manuscript can be evaluated along two lines: a) Insights regarding the accuracy of the self-consistent time-dependent mean field theory as introduced in Refs. [27-33] and applied here. b) Insights regarding the finite time signatures in the quench dynamics first reported in [1].

1) In the abstract the authors state that the mean field approximation is quantitatively accurate at short times and even a good approximation for the thermalization dynamics at late time. Regarding the late time thermalization dynamics this statement is missing an important point: A mean field approximation (time dependent or not) in a translation invariant model cannot lead to a thermal distribution function of moment resolved observables like mode occupation numbers. The authors allude to this when they mention prethermal behavior above Eq. (25), but do not go into detail. A non-expert reader will draw the wrong conclusions from the presentation in this manuscript and believe the mean field analysis is better than it really is: Its shortcomings are known and should be addressed openly.

2) Another question is the accuracy of the mean field approximation when quenching well into the paramagnetic phase. How do time-dependent quantities compare to exact numerical results even for local (non-momentum resolved) observables?

3) It should also be mentioned that the phase diagram of the ANNNO model is considerably more complex if one looks at larger values of kappa/J (see e.g. C. Karrasch and D. Schuricht, Phys. Rev. B 87, 195104). How accurate is the mean field approximation if one starts exploring these parts of the phase diagram? And what about the signatures of the quantum phase transition reported in [1]? In order to make progress regarding this question one also needs to look at other transitions and not only at what was already analyzed using more reliable methods in [1].

In summary, this manuscript is well written, but does not contribute much to a critical analysis of a) and does not add much to b) from the point of view of what was already seen in [1]. Point 1) needs to be addressed before acceptance, points 2) and 3) would be desirable but are not essential.

  • validity: good
  • significance: ok
  • originality: -
  • clarity: -
  • formatting: -
  • grammar: -

Author:  Jacob Robertson  on 2023-04-04  [id 3542]

(in reply to Report 2 on 2023-03-15)

We thank the referee for assessing our manuscript and for providing helpful feedback with which to improve it. The report suggests that our work can be evaluated along two lines:

“a) Insights regarding the accuracy of the self-consistent time-dependent mean field theory as introduced in Refs. [27-33] and applied here.

b) Insights regarding the finite time signatures in the quench dynamics first reported in [1].”

Feature (b) is the motivation and purpose of our work so we will address the reviewer’s comments surrounding this first. Regarding this the referee states our work “does not add much to b) from the point of view of what was already seen in [1].” We do not think that this is a fair assessment. Our work offers several new and important insights that go beyond the work [1] in terms of (i) providing a simple understanding of the cause of the finite time signatures; (ii) clarifying the conditions for their emergence and (iii) establishing a key requirement for them to provide quantitative information about quantum critical behaviour. More precisely:

We point out that a necessary condition for the programme proposed in [1] is that the quantum quench results in an energy density that is small compared to the energy cutoff associated with the critical theory. Our simple theoretical framework readily gives us access to this information, which was not considered in [1]. We observe that for some of the cases considered in Ref [1] this criterion is not well met and the observed signatures are therefore not suitable for extracting any information about the quantum critical point.

By considering two-point functions we establish that the correlation length grows after the quench (as one may have expected) and this provides an explanation for signatures associated with quantum critical scaling. This observation is entirely absent from Ref [1].

By directly working in terms of fermionic variables we show finite time signatures occur also for topological transitions, which provides a significant simplification compared to the discussion in [1].

We explain a peculiar feature visible in the numerical results of [1] as resulting from the formation of a two-particle bound state.

We now turn to criterion (a) mentioned in the report. SCTDMFT and its accuracy in weak-coupling regimes has been previously assessed in considerable detail, for example in past work involving the senior author, cf. Refs [36,37]. The current work concerns a somewhat different situation in that the initial states are such that there is no dynamics without the quench of the interaction parameter. This is closer to the situation considered by Moeckel and Kehrein in Ref.[39]. In this situation the known limitations of SCTDMFT are somewhat less visible and it can provide a quantitatively decent approximation also at late times (as long as we perform a time-average over a short time window). This is all we meant by our comment on the application on SCTDMFT at late times. In hindsight our original discussion was at best unclear and at worst misleading, and we are very grateful for the referee’s comments which enabled us to clarify the why SCTDMFT provides in some sense a decent approximation even at late times. We have now added a discussion of how this is compatible with previous studies of the accuracy of SCTDMFT to replace our previous comments.

We were primarily interested in the short-time regime, where SCTDMFT is expected to be quite good, and this is fully confirmed by comparison with the numerical results of [1]. Another result of our work that we find to be quite interesting is that the mean field theory can capture the persistent oscillations due to the bound state of the exact theory, despite no such bound state existing in the mean field theory, and that moreover within the mean field approximations these oscillations are undamped. In summary we feel that whilst evaluation criterion (a) was not a focus of our work we do in fact contribute something valuable here as well.

Finally, we address the other points raised by the referee. We are of course aware of the fact that the phase diagram of the ANNNI model is quite rich and that there are other phase transitions present. However, we very deliberately restricted our analysis to the parameter regime covered in Ref. [1], because shedding additional light on the findings of that work was our key objective. The mean field approach in terms of Jordan-Wigner fermions is expected to work well for describing the Ising transition for which [1] provided numerical studies to compare its accuracy against. We have not attempted to explore the rest of the phase diagram with this method because there is no a priori reason for the mean-field treatment to be good. Instead, we would advocate using the numerical techniques employed in [1] to study quantum quenches in these regions of the phase diagram.

---

## Editorial Decision

resubmitted